

# Using machine learning to analyze mental health in distance education during the COVID-19 pandemic: an opinion study from university students in Mexico

Roberto Angel Melendez-Armenta, Giovanni Luna Chontal and Sandra Guadalupe Garcia Aburto

División de Estudios de Posgrado e Investigación, Tecnológico Nacional de México/Instituto Tecnológico Superior de Misantla, Misantla, Veracruz, Mexico

## ABSTRACT

In times of lockdown due to the COVID-19 pandemic, it has been detected that some students are unable to dedicate enough time to their education. They present signs of frustration and even apathy towards dropping out of school. In addition, feelings of fear, anxiety, desperation, and depression are now present because society has not yet been able to adapt to the new way of living. Therefore, this article analyzes the feelings that university students of the Instituto Tecnológico Superior de Misantla present when using long distance education tools during COVID-19 pandemic in Mexico. The results suggest that isolation, because of the pandemic situation, generated high levels of anxiety and depression. Moreover, there are connections between feelings generated by lockdown and school performance while using e-learning platforms. The findings of this research reflect the students' feelings, useful information that could lead to the development and implementation of pedagogical strategies that allow improving the students' academic performance results.

# BACKGROUND

The COVID-19 pandemic, declared by the World Health Organization (WHO) in March 2020, has had a profound and transformative impact on many sectors, especially in education (*Karakose, 2021*). The need to mitigate the spread of the virus led to the closure of educational institutions and the massive adoption of distance education modalities (*Bozkurt & Sharma, 2020*; *Odeh & Keshta, 2022*; *Mahasneh et al., 2021*). Globally, students have faced unprecedented challenges due to the abrupt shift to online education. These challenges include not only adapting to new forms of learning but also managing the uncertainty and stress associated with the pandemic (*Marinoni, Van't Land & Jensen, 2020*). In particular, the psychological impact has been considerable, increasing levels of stress, anxiety, and other mental health issues among students (*Qiu et al., 2020*).

In Mexico, the impact has been even more pronounced due to the digital divide. Many students lack access to basic technological tools, which has exacerbated existing inequalities and limited their ability to effectively participate in distance education

Corresponding author
Roberto Angel Melendez-Armenta, ramelendeza@itsm.edu.mx

(*INEGI, 2020*). Distance education, though essential during the pandemic, has presented significant challenges in terms of student stress. Social isolation, the lack of face-to-face interaction with teachers and peers, and uncertainty about the future academic and professional have contributed to increased stress among students (*Aucejo et al., 2020*).

Machine learning (ML) has emerged as a crucial tool for analyzing and responding to the psychological impacts of the pandemic on students. Machine learning algorithms are being used to detect patterns of stress and negative feelings from data collected on educational platforms and online surveys. Studies such as those by *Rahman et al. (2022)* and *Samsari, Mohamad & Selamat (2022)* highlight the use of various machine learning techniques to predict and classify symptoms of stress and depression in students, demonstrating the capability of these methods to provide meaningful and timely analysis that supports psychoeducational interventions.

Factor analysis, a statistical technique used to identify latent variables underlying observed phenomena, has been applied to better understand how various factors contribute to the stress and feelings of students during the pandemic. This allows for a more accurate assessment of students' needs and facilitates the development of more targeted and effective interventions (*Costello & Osborne, 2019*).

# INTRODUCTION

In December 2019 the first cases of a new type of pneumonia in Wuhan, Hubei province, China appeared. The World Health Organization (WHO) described the infection due to the new coronavirus SARS-CoV-2 which produces the COVID-19 disease. Its global expansion has meant a public health crisis with an unprecedented crisis in modern times in accordance with WHO. At the end of January 2020, there were more than 7,000 infections all around the world; the consternation, uncertainty, and different feelings and thoughts took over the population (*WHO, 2020*). In March of that year, *WHO (2020)* had declared COVID-19 as a pandemic, which caused changes in everyday life; each person's way of life experienced a 360 degree turn, and from then on nothing would be the same. This change affected the whole world in different levels; social problems started becoming worse, and deaths were reported daily around the world.

In Mexico, the first COVID-19 case was detected on February 27th, 2020. On April 30th, 64 days later, the number of patients increased exponentially, reaching a total of 19,224 confirmed cases and 1,859 deaths (*Suárez et al., 2020*). In the face of the world sanitary crisis, the Mexican government felt obligated to give a stay-at-home order to the population. For many people, quarantine has been a chaotic experience that changed their everyday activities completely. Moreover, as time passes, days of lockdown have been prolonged, which causes life to be paralyzed within the interiors of each home.

Means of communication inform that a growing number of people feel alone and scared, children suffer because of lockdown, domestic violence is increasing, and many people have lost their jobs and their financial means, and they live in uncertainty. Feelings of fear, anxiety, apathy, indifference, selfishness, desperation, and depression have appeared, because society hasn't been able to adapt to the new way of life yet. Health, safety, economy, and education haven been altered. The latter has been affected as there

was an extreme change in the way in which education processes were carried out. Approximately, more than 850 millions of children and youngsters remain far from schools and universities, with effective national closedown in 102 countries and local closedowns in other 11 (*UNESCO, 2020*). Under these circumstances, teachers and students face new challenges of adaptation, such as working from home and using technological tools within their reach.

In COVID-19 times it has been detected that some students do not dedicate enough time to their studies since they are immersed in different situations that take a toll on them; for example, not being able to understanding the academic topics makes them feel frustration, and even apathy towards dropping out. Apart from this, performance is affected due to the different geographical regions in which students are, because not all of them have the necessary means to take classes, and there are reasons to confirm that the access to technology is not enough (*Krumsvik, 2020*; *Cervantes Holguín, 2020*; *Johnson, Saletti-Cuesta & Tumas, 2020*). In the case of Mexico specifically, in 2019 just 44.6% of homes had a computer as a tool for school support; and 56.4% had internet connection (*INEGI, 2020*). Therefore, this research is designed to answer the following questions:

- RQ1. Is there a relationship between the feelings generated by lock-down and students' behaviours in e-learning platforms required for assessment of mental health?
- RQ2. What are the attributes and machine learning algorithms required for detect stress and feelings effective?

RQ1 will be addressed in factor analysis of the virtual questionnaire applied to students and that allows defining the correlations between items. The identified factors will be used to a new construct for assessment of mental health, and RQ2 will be answered through the implementation of different machine learning algorithms.

In addition, the contributions of this article are:

- A scale to analyze the students' mental health of the Instituto Tecnológico Superior de Misantla during the COVID-19 pandemic in Mexico.
- Machine learning models to detect university students' the stress and feelings of the Instituto Tecnológico Superior de Misantla in distance education.

## RELATED WORKS

The hasty adoption of large-scale teaching, defined as emergency remote learning (*Qiu et al., 2020*; *Priatna et al., 2020*), has enabled some contradictions in the educational systems to come up (*Stewart, 2021*; *Tulaskar & Turunen, 2022*). The most evident are in the denominated high complexity schools in environments of high-risk social exclusion, where it has been confirmed that the social gap hinders digitalization of teaching (*Álvarez et al., 2020*; *Rodríguez, 2020*), even questioning the right to education (*Román et al., 2020*). The most recurring problems in long distance education include homes without WiFi, families with only one mobile device, inadequacy and lack of a living space, a poor cultural background of parents in the formal education field, and communication difficulties with

professors and students. In many of these cases, the course will be lost, and there is even the risk of a long-lasting disconnection, maybe definite, with school (*Vidal et al., 2020*; *Wahyuni, Wulandari & Hardhienata, 2021*; *Duman, 2023*).

Attitudes towards information and communication technologies (ICT) are somewhat uncertain, the development of positive skills and attitudes to work with ICT in both academics and students demands a series of appropriate conditions for its development (*Kalogiannakis, 2007*; *Nag & Roul, 2023*; *Vlachopoulos, 2013*).

At the end of February 2020, when the global alarm because of SARS CoV-2 (COVID-19) began with more intensity, the World Bank established a group of global and multi-sector work to support countries to take necessary measures to face this threat. On April 20, 2020, school close-down were announced, impacting more than 91.3% of the student world population; a total of 1.6 billion, according to *UNESCO (2020)*. As a result of the COVID-19 pandemic, the lives of children, parents and teachers have been altered. Although strategies have been put into practice which allow reducing the impact in education, the great inequality of opportunities that exists amplifies the problem.

In this sense, *UNESCO (2020)* is supporting the application of long-distance learning on a large scale, and it is recommending open educational apps and platforms that schools and teachers can use to reach a larger number of students. Moreover, practices to use mobile low-cost technologies for learning and teaching purposes are being shared to mitigate the distress that education is experimenting in underdeveloped countries.

The virus outbreak and the lockdown could be used as the best way to prove the effectiveness of interventions of educational technology for long distance learning. Unfortunately, few systems are completely prepared. China is one of the countries in which education continued, independently from school's closedown, through internet and distance learning (*Li et al., 2023*). In less prepared countries, the access to technology in most homes can vary, and the access to quality internet or to smartphones is related to the socioeconomic class, even in countries with median income. Therefore, educational politics for the access to distance education must quickly focus on the most needed. Proper planning in the educational systems is something that was learned from the COVID-19 lockdown (*Almusharraf & Khahro, 2020*; *Long & Khoi, 2020*; *Vachkova et al., 2022*). This plan must include an introduction of protocols for evaluations in schools, launch campaigns of hygiene practices, imposition of school close downs, and the distance learning offer. In this sense, the educational measures during a crisis can help to prevent and recover public health, at the same time that they reduce the impact in students and their learning. Moreover, it is important to point out that education has the potential to contribute to children and youth protection (*Bozkurt & Sharma, 2020*; *Wang et al., 2020*).

In regard to higher education, the United Nations Educational, Scientific and Cultural Organization (UNESCO) has planned a series of recommendations in which it is mentioned that it is urgent that governments and institutions of those countries where the epidemic is starting to manifest, and that they plan proper measures to protect the health of citizens avoiding the violation of the right to education (*UNESCO, 2020*). Moreover, the educational authorities must supervise that these measures are put into practice to favor the fact that students continue to learn in spite of the temporary closedown of the

institutions. In this sense, *UNESCO (2020)* recommends higher education institutions to take the following measures:

- Spread among the university community information and recommendations only and exclusively from sanitary national authorities and the WHO to avoid alarmism and the spread of rumors or fake news (*UNESCO, 2020*).
- Regularly use their own website and social networks so that the university community is informed on time and truthfully about COVID-19, including recommendations on behaviors to follow preventive or in case of infection, as well as the last advances in investigation counteracting in an active way racist or discriminatory attitudes and behaviors that could emerge as a result of distortions (*UNESCO, 2020*).
- Attend to the instructions and recommendations from national authorities and actively participate in the mechanisms of inter-university coordination to unfold in a coordinate and coherent way at national scale (*UNESCO, 2020*).
- The use of the university's own platform for online learning, or the virtual campus, to continue to facilitate distance learning to students. In the case that the university does not have one, install one of the many open educational apps and platforms, taking into account that some of the students might only be able to use mobile devices (*UNESCO, 2020*).

This will demand having, in turn, mechanisms of training and support online for professors and students that should be correctly reinforced. This information is being updated and can be verified through COVID-19 infection maps (*Urzúa et al., 2020*; *Adhikari et al., 2020*).

In addition, the use of learning management systems (LMS), has grown considerably in universities around the world, which leads students to use their mobile devices; however, students find this platform as a repository for electronic documents and not as a learning tool (*Papadakis et al., 2018*).

## MATERIALS AND METHODS

In this research a survey was used to measure the feelings that students presented when using technological tools during the COVID-19 pandemic. The survey provides a quantitative understanding of the perceptions of university students in Mexico in terms of education and its relationship with the world pandemic. In addition, a factor analysis has been performed to measure the reliability of the construct and different machine learning algorithms were applied to detect university students' feelings in distance education. Exploratory faction analysis (EFA) begins with the assumption that any observed variable can be linearly related to some underlying factors. The process involves extracting factors based on their eigenvalues, deciding the number of factors to retain through criteria such as Kaiser's criterion or the scree test, and then rotating the factors to achieve a simpler, more interpretable structure (*Steiner & Grieder, 2020*). Various rotation techniques, such as Varimax (orthogonal) and Promax (oblique), can be applied depending on whether the factors are assumed to be correlated or not (*Costello & Osborne, 2019*).

Figure 1 shows the research methodology used which consists of three main phases that are described below.

## Participants

The participants of the opinion study were a group of undergraduate students from different university degrees from the Instituto Tecnológico Superior de Misantla during the January to July 2021 semester. The survey to university students was carried out based on a simple random sample, providing a sample size of $n = 381$ participants.

It is important to mention that the Instituto Tecnológico Superior de Misantla granted Ethical approval to conduct the study within its facilities, which was established by the Institutional Review Board. Additionally, written informed consent was obtained from the study participants.

## Proposed scale

To identify the feelings and the perception of learning in online education during the COVID-19 pandemic, the participants completed a virtual questionnaire using the free-use tool Google Forms. The survey was divided in three groups of questions: (1) general information, (2) feelings in regard to long distance learning, and (3) pandemic future. The general questions included age, sex and degree. Regarding the second group of questions, the candidates were asked to answer to topics related to feelings towards the new type of learning (e-learning). The last part include one question about their educational future.

**Part 1** The first group of questions were related to student' general information.

- Degree: Which is your engineering degree?
- Age: How old are you?
- Gender: Which is your gender?

**Part 2** The second group of questions applied to students can be observed below:

- Q1: Did I feel attention, commitment and empathy from my teachers during the time they worked online?
- Q2: Did I feel too much workload in the activities in each of my subjects?
- Q3: Did the activities sent in your subjects overwhelmed, tired or frustrated you emotionally or physically?
- Q4: Did you feel positive during your online learning? For examples, safe, satisfied, happy, relaxed and/or pleased
- Q5: Did you concentrate taking online classes?
- Q6: Did I feel stressed because of my time distribution when taking online classes?
- Q7: Did I feel that I was unfairly asked to continue or adapt my academic duties in the presence of the pandemic?
- Q8: Did you consider that your school performance during online classes accomplished what you were hoping for?
- Q9: Did you feel that COVID-19 psychologically affected your school performance?

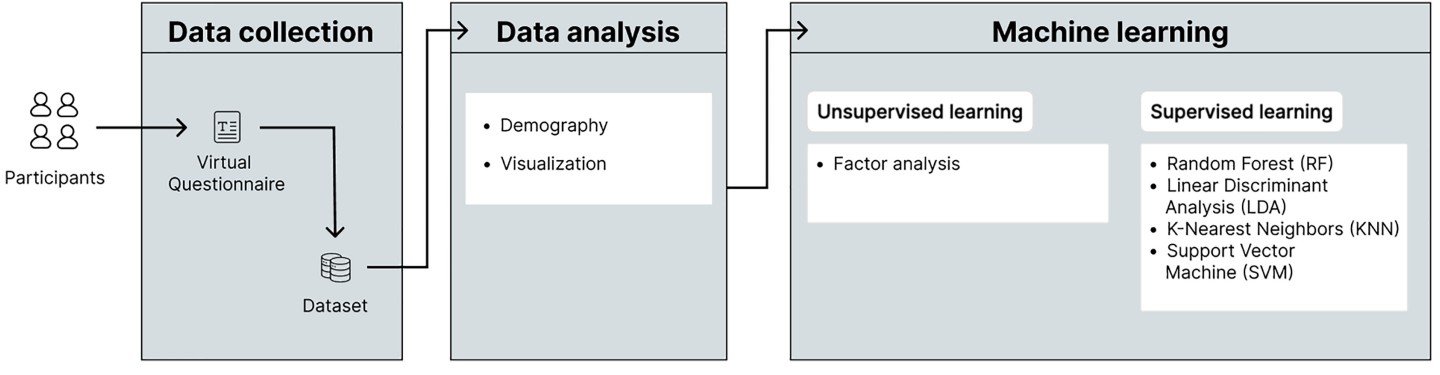

**Figure 1  Proposed research methodology.**               

Table 1 shows frequencies of second group of questions in virtual questionnaire answered on a Likert scale, where 1 = Never, 2 = Sometimes or 3 = Always with each question.

**Part 3** The last question (LQ) measures feelings of university students about returning to online classes.

• LQ: What would you feel if you were informed that classes would be online again?

Table 2 shows frequencies of last question in virtual questionnaire answered on a Likert scale, where 1 = Joy, 2 = Stress or 3 = Uncertainty.

The questionnaire was applied and is originally in Spanish. Lastly, the names of participants were not collected with the objective of protecting their integrity.

## Exploratory factor analysis

EFA is a statistical method used to uncover the underlying structure of a set of observed variables (*Yong & Pearce, 2013*). It helps identify latent constructs that cannot be measured directly by analyzing patterns of correlations or covariances among variables. EFA is particularly useful in the early stages of research to simplify data, reduce dimensionality, and formulate hypotheses for future studies (*Williams, Onsman & Brown, 2010*).

EFA begins with the assumption that any observed variable can be linearly related to some underlying factors. The process involves extracting factors based on their eigenvalues, deciding the number of factors to retain through criteria such as Kaiser's criterion or the scree test, and then rotating the factors to achieve a simpler, more interpretable structure (*Steiner & Grieder, 2020*). Various rotation techniques, such as Varimax (orthogonal) and Promax (oblique), can be applied depending on whether the factors are assumed to be correlated or not (*Costello & Osborne, 2019*).

This technique has been widely applied in psychology, education, and social sciences to explore potential dimensions of constructs measured by questionnaires or tests. Through EFA, researchers can gain insights into the dimensions that underlie a set of variables, guiding the development of more precise instruments and contributing to theory building (*Henson & Roberts, 2006*).

**Table 1 Frequencies of second group of questions answered on a Likert scale.**

| Questions | 1 | 2 | 3 |
|-----------|-----|-----|-----|
| Q1 | 1.05% | 51.71% | 47.24% |
| Q2 | 6.82% | 69.29% | 23.88% |
| Q3 | 15.75% | 63.25% | 20.99% |
| Q4 | 18.64% | 64.30% | 17.06% |
| Q5 | 13.65% | 63.52% | 22.83% |
| Q6 | 18.63% | 55.38% | 25.98% |
| Q7 | 76.12% | 0.00% | 23.88% |
| Q8 | 66.67% | 0.00% | 33.33% |
| Q9 | 27.30% | 11.81% | 60.89% |

**Table 2 Frequencies of last question answered on a Likert scale.**

| Questions | 1 | 2 | 3 |
|-----------|-----|-----|-----|
| LQ | 6.04% | 65.09% | 28.87% |

## Confirmatory factor analysis

Confirmatory factor analysis (CFA) is a statistical technique used to test the hypothesis that the relationships among a set of observed variables can be explained by a smaller number of underlying latent variables (factors) (*Thompson, 2004*). Unlike exploratory factor analysis (EFA), which is used to identify potential underlying structures, CFA is used to verify the extent to which a hypothesized factor model fits the observed data. This approach allows researchers to test specific theories and models of factor structure that have been proposed based on previous research or theoretical considerations.

CFA involves specifying a model, estimating the parameters of the model, and then evaluating the fit of the model to the observed data (*Brown, 2015*). It requires *a priori* specification of the number of factors and the pattern of loadings of observed variables on these factors. Model fit is assessed using various indices, such as the Comparative Fit Index (CFI), the Tucker-Lewis Index (TLI), and the Root Mean Square Error of Approximation (RMSEA) (*Alavi et al., 2020*).

This method is widely used in psychology, education, and social sciences for construct validation, measurement invariance testing, and scale development. CFA has become an essential tool in the validation process of measurement instruments, allowing researchers to refine scales and assess their reliability and validity in measuring theoretical constructs (*Marsh et al., 2014*).

## Evaluation metrics

A confusion matrix is a powerful tool used in machine learning to evaluate the performance of classification models, offering a detailed breakdown of prediction results in a tabular format (*Amin & Mahmoud, 2022*; *Krstinić et al., 2020*). It categorizes predictions

into true positives (TP), true negatives (TN), false positives (FP), and false negatives (FN), allowing for a nuanced analysis of a model's accuracy (*Heydarian, Doyle & Samavi, 2022*). True positives and true negatives indicate correct predictions for positive and negative classes, respectively, while false positives and false negatives highlight errors in prediction (*Görtler et al., 2022*).

Several performance metrics can be derived from the confusion matrix, including accuracy, precision, recall (sensitivity), and the F1 score (*Zeng, 2020*). Accuracy measures the overall correctness of the model, precision evaluates the model's ability to identify relevant instances correctly, and recall assesses the model's capability to find all relevant instances (*Pommé et al., 2022*). The F1 score balances precision and recall, providing a single metric to gauge model performance, especially in imbalanced datasets (*Vanacore, Pellegrino & Ciardiello, 2022*).

The confusion matrix not only reveals a model's strengths and weaknesses but also guides improvements by pinpointing areas where the model confuses classes (*Valero-Carreras, Alcaraz & Landete, 2023*). It is essential for validating the efficacy of classification algorithms, facilitating the refinement of models for better reliability and validity in theoretical construct measurement (*Krstinić et al., 2020*).

The performance of the model is assessed using accuracy, recall, precision, F-score, and the confusion matrix.

$$\text{Accuracy} = \frac{(TN + TP)}{(TP + TN + FP + FN)} \tag{1}$$

$$\text{Precision} = \frac{TP}{(TP + FP)} \tag{2}$$

$$\text{Sensitivity/Recall} = \frac{TP}{(TP + FN)} \tag{3}$$

$$F-\text{score} = 2 \times \frac{\text{Precision} \times \text{Recall}}{\text{Precision} + \text{Recall}} \tag{4}$$

## RESULTS

### Participants' demography

The breakdown by gender, age and degree in the students (st) that answered the survey is shown in Table 3. In this questionnaire students' gender considered feminine, masculine and other, nevertheless, all the answers were under the categories of masculine and feminine. The survey was open to the nine engineering degrees that the Instituto Tecnológico Superior de Misantla offers, but the only participants were from industrial engineering (IE), civil engineering (CI), electro mechanic engineering (EME), computer systems engineering (CSE), company management engineering (CME), environmental engineering (EE), and communications and information technologies engineering (CITE).

From the 381 students who answered the form, approximately 52.75% were women and 47.26% (st = 180) were men. The age average was 20 years old. Students with more participation are from the industrial engineering degree, 50.13% (st = 191), followed by civil engineering, 17.58% (st = 67), and electro mechanic engineering, 12.33% (st = 47).

**Table 3 Participants' information by degree, age, and gender.**

| Engineering degrees | N | Mean (age) | SD (age) | Male (%) | Female (%) |
|---|---|---|---|---|---|
| IE | 191 | 20.51 | 1.23 | 30.36 | 69.64 |
| CE | 67 | 19.85 | 1.36 | 67.17 | 32.83 |
| EME | 47 | 19.55 | 0.83 | 80.86 | 19.14 |
| CSE | 37 | 21.56 | 1.55 | 70.28 | 29.72 |
| CME | 23 | 20.87 | 0.75 | 17.39 | 82.61 |
| EE | 8 | 19.12 | 0.99 | 37.5 | 62.5 |
| CITE | 8 | 20.62 | 1.5 | 75 | 25 |

The degrees with less participation were environmental engineering and communications and information technologies engineering, 2% (st = 8) each one.

## Visualization

The analysis of the first six questions showed uncertainty in students when receiving online education, due to a higher frequency in the answer "sometimes" in relation to "never" and "always". Frustration (>50%) and stress (>50%) are the symptoms that are regularly present in students in relation to the results observed in Fig. 2. Moreover, students indicated a lack of concentration to carry out the academic activities at home during the period of contingency due to the COVID-19 pandemic. In this sense, the students indicated that sometimes (69%) or always (24%) felt too much workload in their academic tasks.

## Factors

The exploratory factor analysis allowed to find the patterns of correlation among the questions (items) answered in the survey conducted with university students. A value of 0.83 was obtained in the Kaiser-Meyer-Olkin (KMO) test, indicating that the data are into "meritorious" category according to *Kaiser & Rice (1974)*. Additionally, the results of Bartlett's test was found to be significant. Therefore, the data pertaining to the proposed scale in this research were deemed suitable for conducting exploratory factor analysis.

The number of factors necessary to adequately represent the covariance structure of the observed variables was determined. The first three factors have eigenvalues of 3.17, 1.13, and 0.96, respectively. Kaiser's criterion suggests retaining factors with eigenvalues >1 (*Kaiser, 1960*), and for that reason two significant factors are identified. Although the third factor has an eigenvalue slightly below 1, its proximity suggests that it could be relevant depending on the context of the analyzed variables.

The eigenvalues and variance indicate that the first two factors have eigenvalues >1 (3.17 and 1.13, respectively), justifying their retention according to Kaiser's criterion. These two factors explain 15.64% and 12.76% of the total variance.

Afterwards, factor loadings were analyzed to indicate the association between each variable and the identified factors. Variables with high loadings on the first factor may be

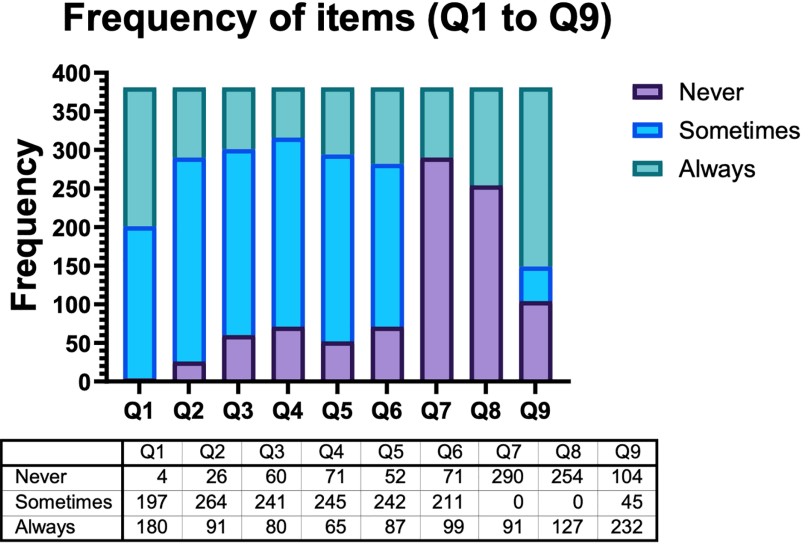

**Figure 2  Frequency of variables Q1 to Q9.**

related to each other, representing a common construct, while the same applies to variables with high loadings on the second factor. Table 4 shows how the variables were associated with any of the two factors, *i.e.*, factor loadings ≥0.3.

The interpretation of factor loadings is based on:

- Variables Q2, Q3, Q6, and Q7 have the highest loadings on the first factor, indicating they share a common dimension.
- Variables Q4, Q5, and Q8 show the highest loadings on the second factor, suggesting another distinct underlying construct that these variables represent.
- Variable Q1 exhibits significant loading on both factors, implying they may have a complex relationship with the constructs represented by these two factors. Therefore, it is not assigned to either factor.
- Variable Q9 does not score the minimum factor loading requirement of 0.3.

Therefore, the factor loadings for each item are defined as follows:

- Factor 1: **Academic load and pandemic stress** (Q2, Q3, Q6, and Q7). This sub-scale represents students' overall perception of academic workload, emotional and physical stress experienced due to tasks and time distribution during online classes, as well as the perception of injustice or additional challenges related to the pandemic.
- Factor 2: **Online learning experience** (Q4, Q5, and Q8). This sub-scale reflects the relationship between positive responses during online learning, concentration during classes, and satisfaction with the academic performance achieved during that period. This factor would provide valuable information about the effectiveness and impact of the online learning environment, student satisfaction and well-being.

**Table 4 Factor loadings associated with each identified factor.**

| Item | Factor 1 | Factor 2 |
|------|:--------:|:--------:|
| Q1 | **−0.38** | **0.38** |
| Q2 | 0.61 | |
| Q3 | 0.71 | |
| Q4 | | 0.59 |
| Q5 | | 0.63 |
| Q6 | 0.48 | |
| Q7 | 0.35 | |
| Q8 | | 0.58 |
| Q9 | **0.25** | |

## Dimensions

An exploratory graph analysis (EGA) was conducted for the variables defined in EFA and assess the number of dimensions; the dimensions in the graph are considered latent variables or factors. Figure 3 is a network diagram with two clusters, and red lines representing negative correlations and green lines representing positive correlations between variables, with the thickness of the lines indicating the strength of the correlation.

As expected, the first cluster (dimension) consists of variables Q2, Q3, and Q4, while the second cluster consists of items Q4, Q5, and Q8. Analyzing the graph, strong positive relationships (green) between Q4 and Q5 are observed, as well as negative relationships (red) between Q2 and Q4.

In addition, CFA was conducted to test the model proposed by the EGA network and results allowed for the relationship between the observed variables, Q2, Q3, Q6, Q7, Q4, Q5, Q8, and latent variables, factor 1 and factor 2.

The obtained value $\chi^2 = 11.412$ with a $p-value = 0.576$. Since the $p-value > 0.05$, there is not enough evidence to reject the null hypothesis, indicating that the model fits the data well. $RMSEA = 0.000$ with a 90% confidence interval ranging from 0.000 to 0.045 and an associated $p-value = 0.970$ for the $RMSEA <= 0.05$ test. Therefore, this value indicates a good fit in the model.

Factor loadings for Factor 1 (see Table 5) and Factor 2 (see Table 6) are all statistically significant, indicating that all observed variables contribute significantly to their respective factors. For example, Q3 has a factor loading of 1.295 on F1, suggesting a strong positive relationship with that latent variable. Besides, the covariance estimate between factor 1 and factor 2 is −0.084 with a $p-value = 0.000$, indicating a significant negative relationship between the two factors. However, the magnitude of this relationship is small. The negative relationship between the two factors suggests that academic load and pandemic stress decreases as online learning experience increases.

Furthermore, Tables 5 and 6 show $p-values < 0.001$, which reflects robust statistical significance in the associations between the observed variables and the latent factors. This uniform pattern indicates that the variables are not only significant predictors of the latent

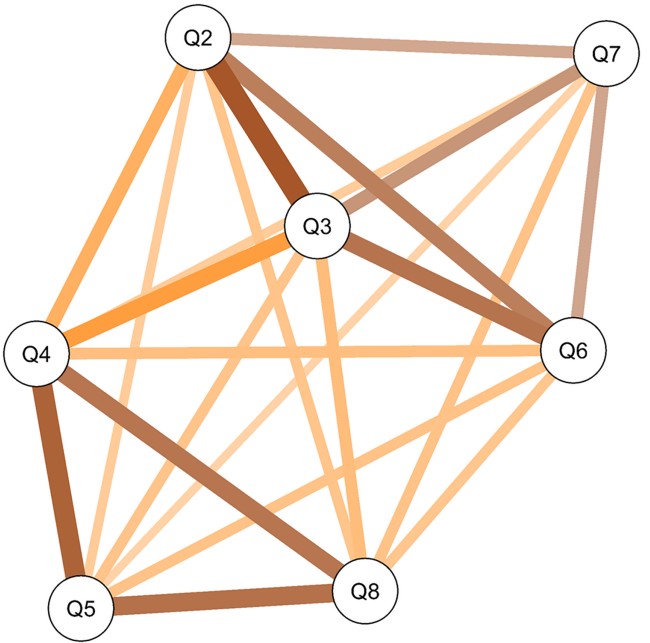

**Figure 3** EGA network with two dimensions calculated from the data of the proposed scale.

**Table 5 Statistical values of Factor 1 obtained by CFA.**

| Item | Estimate | Std-error | $z - value$ | $p - value$ |
|------|----------|-----------|-------------|-------------|
| Q2 | 1.000 | | | |
| Q3 | 1.295 | 0.150 | 8.652 | <0.001 |
| Q6 | 1.098 | 0.142 | 7.749 | <0.001 |
| Q7 | 1.070 | 0.170 | 6.284 | <0.001 |

**Table 6 Statistical values of Factor 2 obtained by CFA.**

| Item | Estimate | Std-error | $z - value$ | $p - value$ |
|------|----------|-----------|-------------|-------------|
| Q4 | 1.000 | | | |
| Q5 | 0.882 | 0.111 | 7.981 | <0.001 |
| Q8 | 1.316 | 0.169 | 7.793 | <0.001 |

factors but also that the probability of these associations being due to chance is extremely low, reinforcing the robustness of our confirmatory factor model.

Figure 4 shows the resulting path diagram from the CFA, where the nodes labeled as F1 and F2 represent the latent factors. Each factor is associated with certain observed variables (Q2, Q3, Q6, Q7 for F1 and Q4, Q5, Q8 for F2), which are represented by rectangles. The solid arrows between the factors and the variables indicate the factor loadings, with the numbers above the arrows representing the standardized coefficients of the factor loadings.

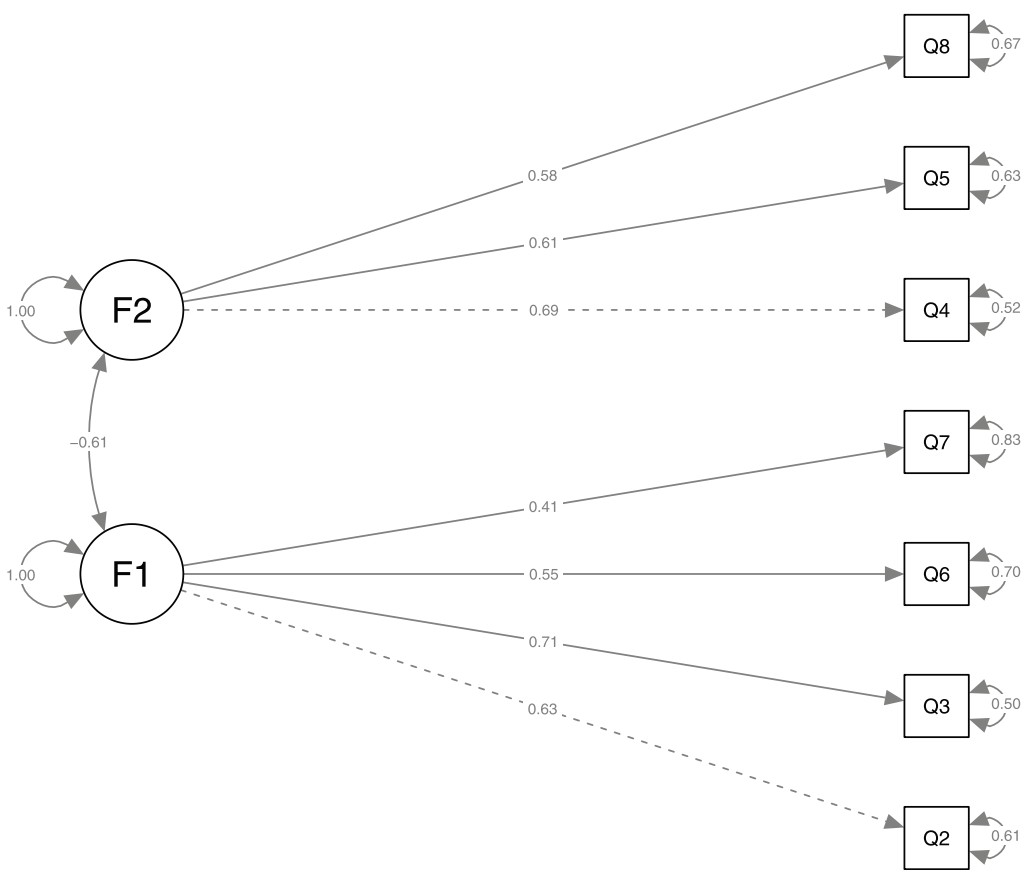

**Figure 4 Path diagram of proposed scale to calculate the fit of the EGA model.**

## Supervised learning algorithms

Four machine learning models were used to identify university students' feelings about returning to online classes. According to the results of the factor analysis, Q2, Q3, Q6, Q7, Q4, Q5, and Q8 were defined as input variables, and LQ was used as output variable (class). The configuration used for the model hyper-parameters was to the following: The interpretation of factor loadings is based on:

- **Random Forest (RF)**. Set at 100 to define the number of trees in the forest, and random state, fixed at 42 for result reproducibility, were used.
- **Linear discriminant analysis (LDA)**. Hyper-parameters such as solver were utilized to select the calculation algorithm, and for simplicity, the number of components was kept at 1.
- **K-nearest neighbor (KNN)**. It mainly utilized five neighbors, which specifies that five nearest neighbors were considered for a point's class voting. This value directly influenced the classification decision, with a smaller number made the model more noise-sensitive, and a larger number potentially smoothing decision boundary.

**Table 7 Performance metrics of the machine learning models.**

| Model | Precision | Accuracy | Recall | F-score |
|---|---|---|---|---|
| RF | 0.95 | 0.92 | 0.96 | 0.95 |
| LDA | 0.95 | 0.95 | 1 | 0.97 |
| KNN | 0.95 | 0.87 | 0.91 | 0.92 |
| SVM | 1 | 0.95 | 1 | 1 |

- **Support vector machine (SVM)**. The linear kernel was used to define the type of hyperplane used to separate the classes. This type of kernel computes the dot product of the two vectors and adds an optional constant.

All models were implemented in the Python programming language, and data were split into 70% training and 30% test, split performed randomly. Besides, the random_state parameter was defined, which sets a seed for the random number generator, allowing the training function of the model to be reproduced.

Table 7 shows the comparison of performance metrics (precision, accuracy, recall, and F-score) of the test set obtained by the machine learning models: RF, LDA, KNN, and SVM.

In this research, what interests us is that the models correctly classify feelings of university students into joy, stress, and uncertainty. It is clear from the table that the SVM model is the best classifier for feelings of university students regarding the return to online classes, with a precision of 100%. Additionally, it can be said that classification yields good results when using the RF model, as it has the following percentages: precision 95%, accuracy 92%, recall 96%, and F-score 95%.

## DISCUSSION

The findings of this research underscore significant contributions to the field of distance learning and student mental health in the context of the COVID-19 pandemic. *Qiu et al. (2020)*, *Priatna et al. (2020)*, *Rodríguez (2020)* have addressed the rapid adoption of distance education, highlighting the challenges and contradictions in educational systems, particularly in contexts of high social exclusion. These studies agree that the social divide negatively impacts the digitalization of teaching, a critical point also identified by *Vidal et al. (2020)*, who emphasizes the risk of a prolonged and possibly definitive disconnection from the educational system for some students.

The current research complements these findings by investigating the psychological impact of confinement on students, using machine learning algorithms to classify students' feelings about online learning. This innovative approach not only corroborates previous studies' observations of the academic and emotional challenges faced by students but also offers specific technological solutions for their identification and management. The results show that stress and anxiety are prevalent among students due to confinement, and there is a clear relationship between these feelings and academic performance on e-learning

platforms, aligning with concerns raised by *Álvarez et al. (2020)*, *Bozkurt & Sharma (2020)* regarding the effects of the pandemic on education and mental health.

The use of ML techniques during the COVID-19 pandemic has proven to be indispensable not only for addressing educational and mental health challenges but also for informing policy and strategic decisions aimed at improving educational systems, especially in developing countries. The integration of ML allows not only the detection of patterns of stress and negative feelings in students, as observed in studies by *Rahman et al. (2022)*, *Samsari, Mohamad & Selamat (2022)*, but also provides guidelines for precise and evidence-based interventions.

We hope that these findings will be utilized by the scientific community and policymakers to develop more resilient and adaptive educational policies that can respond to emerging needs during similar crises in the future. In particular, ML models can help quickly identify areas of greatest need and distribute resources more effectively, which is crucial for developing countries with limited resources. Furthermore, previous studies have shown similar correlations between student stress and contributing factors during the pandemic, as indicated by research from *Qiu et al. (2020)*, *Aucejo et al. (2020)*, underscoring the consistency and relevance of our current findings. These studies not only validate our conclusions but also reinforce the need for a data-driven approach to understanding and mitigating the psychological and educational impacts of the pandemic.

Other efforts to use ML in similar contexts include work by *Aggarwal, Girdhar & Alpana (2022)* and *Naiem et al. (2022)*, who have applied predictive models and sentiment analysis to assess the effectiveness of e-learning tools and to detect psychological support needs among students. These applications demonstrate the versatility and potential of ML to be adapted to various educational needs and contexts, offering a robust framework for future research and practical applications. With the application of these advanced techniques, we can not only better understand the challenges faced by students but also better anticipate and prepare educational institutions to face future crises, ensuring continuous and quality education despite adversities.

## CONCLUSIONS

The present study set out to explore the relationship between confinement and changes in students' psychological well-being, using e-learning platforms as tools to assess stress and negative feelings. The research focused on two main research questions: the first sought to establish whether there is a direct correlation between the feelings generated by confinement and behaviors observed on online learning platforms; the second focused on determining the most effective machine learning algorithms and attributes to identify signs of stress and emotional distress.

To address these questions, the study used a combination of quantitative and qualitative analysis, including monitoring students' interaction with e-learning platforms and the application of advanced machine learning models to analyze data on students' behavior and feelings. The findings confirmed that confinement has significantly exacerbated stress levels among students, affecting their academic performance and emotional well-being. The results presented here are based solely on self-report data. As such, they are inherently

limited by the characteristics of this type of data, including self-selection bias and the potential for respondents to describe their experiences and feelings inaccurately or incompletely.

Furthermore, several machine learning algorithms were identified as promising tools for effectively detecting stress and negative feelings, thus offering avenues for early and personalized interventions. While the machine learning models demonstrated strong performance without requiring adjustments to additional parameters, it is important to acknowledge that the results might be influenced by factors not considered in this study. Future research should investigate additional variables that could impact the relationship between isolation and stress, including social support, personal resilience, or specific conditions relevant to the context of Mexican universities.

This study achieves its overall objective by demonstrating through empirical evidence how confinement has impacted students' mental health. Using a rigorous methodological approach along with machine learning technologies, new possibilities are opened for monitoring and intervening in mental health within educational environments. The relevance of adopting adaptive and responsive strategies to promote student well-being in crisis situations is emphasized. Consequently, the conclusions of this analysis suggest the need to focus more on the psychological facets associated with virtual learning, proposing to effectively integrate educational technological resources with emotional support, marking a path forward to improve support for students.

### Funding
The authors received no funding for this work.

### Competing Interests
The authors declare that they have no competing interests.

### Author Contributions
- Roberto Angel Melendez-Armenta conceived and designed the experiments, performed the experiments, analyzed the data, performed the computation work, prepared figures and/or tables, authored or reviewed drafts of the article, and approved the final draft.
- Giovanni Luna Chontal conceived and designed the experiments, performed the computation work, authored or reviewed drafts of the article, and approved the final draft.
- Sandra Guadalupe Garcia Aburto conceived and designed the experiments, authored or reviewed drafts of the article, and approved the final draft.

### Ethics
The following information was supplied relating to ethical approvals (*i.e.*, approving body and any reference numbers):

The Instituto Tecnológico Superior de Misantla granted Ethical approval to carry out the study within its facilities (Ethical Application: ACTA DE 2da. SESIÓN EXTRAORDINARIA).

## Data Availability

The raw data and code are available in the Supplemental Files.

## Supplemental Information

Supplemental information for this article can be found online at http://dx.doi.org/10.7717/peerj-cs.2241#supplemental-information.

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
