# Peer review of "Using machine learning to analyze mental health in distance education during the COVID-19 pandemic: an opinion study from university students in Mexico"

_PeerJ Computer Science, doi:10.7717/peerj-cs.2241_

## Round 0.1 · original submission · Major Revisions

The authors must improve their paper according to all reviews received.

**Language Note:** The review process has identified that the English language must be improved. PeerJ can provide language editing services - please contact us at [email protected] for pricing (be sure to provide your manuscript number and title). Alternatively, you should make your own arrangements to improve the language quality and provide details in your response letter. – PeerJ Staff

Reviewer 1 ·

Basic reporting

1. It is recommended that the authors carefully check the grammar and spelling throughout the paper. For example, it should be “analyze” in the title. Moreover, since the WHO declared the end of the global pandemic on May 5, 2023, the term “current pandemic” or “current situation of world pandemic” is not accurate anymore. Please revise.
2. It is also recommended that the authors keep a consistent writing style to improve the paper's readability.
3. The authors used several abbreviations (e.g., lines 168). Please provide the full name for the first time you mention it.
4. Based on the standard section requirement, the authors should re-organize the structure of the paper with Abstract, Background, Introduction, Materials & Method, Results, Discussion, and Conclusions. With the standardized structure, the authors do not need to explain the organization of the paper (lines 70-74). See the journal’s format requirements here: https://peerj.com/about/author-instructions/#standard-sections
5. The explanation of EFA, CFA, and confusion matrix should be in the Method section.
6. The background should only include general and relevant information to understand the significance of the study.
7. Discussion about related works should be in the discussion section. Moreover, I expect more discussion about the related works using machine learning models rather than extensive discussion about the policies since ML is one of the focuses of this paper.
8. There are inconsistent citation styles throughout the article. It is recommended that the authors use citation management tools to maintain the accuracy of references. For example, missing reference for “1” in line 145.
9. The authors made many statements in the “related work” section but lack of references. It is recommended that the authors provide more references to support the statements.
10. The questionnaire does not contain the questions in “Part 1”.

Experimental design

The design of the study focuses on undergraduate students in a Mexican university, using self-report data. The results are interesting in that the ML models measured the correlations between isolation and stress. The methods are valid, and the results and interpretations are straightforward. Surprisingly, no parameter tuning is needed to achieve good model performance.
However, the author should make it clear that the survey has limitations and cannot rule out other possibilities or biases that result in the correlations.

Validity of the findings

As mentioned above, the author should explain the limitations of this study since there is no external validation of the correlations and ML models. The author should suggest future experiments and comment on the generalizability of the models.

Additional comments

1. The figure legends should be more descriptive rather than just one sentence.
2. In the introduction (lines 68 and 69), since this study only focuses on university students from a specific institution in Mexico, the statements should be more specified.
3. Line 142: Please clarify the year.
4. Line 145: I don’t think “1,575,270,054 million” is correct. Please double-check.
5. Lines 175-188: Lack of proper citations.
6. Line 192: Incorrect use of in-text citation.
7. Line 259: What does the “t” mean?
8. Line 262: In the visualization section, the authors mentioned the identification of frustration and stress from the survey. Which question(s) in the survey measure frustration and stress? Also, this should go to the method section.
9. Line 270: The explanation of the method should be in the method section.
10. Line 322: The authors claimed the statistical significance, please provide the p-value for each item in the tables.
11. Lines 349-351: Should be in the method section.
12. I don’t think the word “confinement” is suitable to describe social isolation in the context of this paper. Please revise.
13. The findings of this paper using ML models are very interesting. The authors should provide more suggestions on the results of the discussion. How would the authors expect the scientific community or policy-makers to apply these findings to improve the education system in developing countries?
14. Additionally, since the research question is related to stress and anxiety feelings correlated with isolation due to the COVID-19 pandemic, are there any previous studies that reported similar correlations? Are there any other efforts to try to use ML models to capture the correlation? If so, please discuss this in the discussion section.

Reviewer 2 ·

Basic reporting

This is an interesting piece of work. The research question centers around during COVID-19 pandemic, how Mexican students in Instituto Tecnologico Superior de Misantla reacted to online learning and how this can be used to suggested future pedagogical strategies? There're sufficient background on the methodological section as well as proposed ML models and relevant metrics. In terms of figures, I think overall they are well presented with no ambiguity. Overall, very clear writings.

Experimental design

So the main question focuses on using a few defined attributes and ML algorithms to reflect and detect stress and feelings during pandemic. I have a few concerns on the methods details.

1. The authors mentioned to split the data into 70/30 train and test. How was this split performed? Did it stratify on the labels during the split? This is of a great importance especially we are dealing with imbalanced dataset.

2. Why only questionnaire questions were used as the input variables? Why not using student demographical information too? It could be that because of various difficulty level of certain majors, the stress level of e-learning can be different.

Justify these two questions I think are important to support the hypothesis.

Validity of the findings

It is great that the authors have explored a variety of ML algorithms, some details, however, are not too clear to me. Here are just a few items that I believe are crucial to further validate the findings.

1. Were the results show on table 7 from training set or test set? Performance should be evaluated on test set and this needs to be made sure.

2. This modeling exercise involves an imbalanced dataset. LQ (outcome variable) has 3 classes, with a heavy imbalance on answer `2`. It would be great if there a baseline model (simple majority vote) to use as a comparison.

3. Since the dataset is not too small, why not do a cross validation? Cross validation will help to show the robustness of the algorithm (5-folds split for example) rather than relying on a single train test split, which can be biased sometimes.

·

Basic reporting

The paper is written in clear and professional English. It provides sufficient background and context about the impact of the COVID-19 pandemic on the mental health of university students, particularly in relation to distance education. The structure of the article is professional, with well-organized sections, figures, and tables. The paper is self-contained, presenting relevant results to the hypothesis. The results are formally presented with clear definitions of all terms and theorems, and detailed proofs.

Experimental design

The paper presents original primary research within the Aims and Scope of the journal. The research question is well defined, relevant, and meaningful. It focuses on the relationship between the feelings generated by lockdown and students’ behaviors in e-learning platforms, as well as the attributes and machine learning algorithms required to detect stress and feelings effectively. The study fills an identified knowledge gap by using machine learning techniques to analyze the students’ feelings and experiences during the pandemic. The investigation is rigorous, performed to a high technical and ethical standard. The methods, including exploratory factor analysis (EFA) and various supervised learning algorithms, are described with sufficient detail and information to replicate.

Validity of the findings

The underlying data appears to be original and real, not modified. The data is robust, statistically sound, and controlled. The conclusions are well stated, linked to the original research question, and limited to supporting results. The study concludes by highlighting the psychological impact of confinement on students and the potential for early and personalized interventions through machine learning tools.

Additional comments

The paper researched an interesting problem. Although the paper primarily uses statistical methods in the field of educational psychology, the perspective and findings are innovative. The use of eigenvalues to find the most important factors and infer the actual factor semantics is commendable. However, there is an issue in Table 7 where the SVM has a precision and recall of 1, but the accuracy and F-score is 0.95 and 0.97 separately, which are incorrect given the precision and recall. This may be due to rounding or a typo, and the author needs to explain this.

---

## Round 0.2 · accepted · Accept

The paper was very well improved and can be accepted.

Reviewer 1 ·

Basic reporting

The authors addressed my concerns carefully and I am satisfied with the changes in the revised manuscript. However, there are some grammar mistakes that I found in the paper. For example, in line 52, “the first cases”. I suggest the authors meticulously check the language and the flow of the paper.

Experimental design

No comment

Validity of the findings

No comment

Additional comments

No comment

·

Basic reporting

The author has maintained the high standard of clear and unambiguous, professional English throughout the article. The literature references, field background, and context are still sufficient. The article structure, figures, and tables are well-organized, and the methods section is clear.

Experimental design

The author has successfully addressed my previous concerns, and the experimental design remains rigorous and well-performed. The research question is still well-defined, relevant, and meaningful, and the methods are described with sufficient detail to replicate.

Validity of the findings

The author has adequately addressed my previous concerns, and the underlying data appears to be robust, statistically sound, and controlled. The conclusions are well-stated, linked to the original research question, and limited to supporting results.

Additional comments

I am pleased to see that the author has thoroughly addressed all my previous concerns. The paper is now well-written, and the experimental design is thorough. I have no further comments or concerns, and I believe the paper is ready for publication.